# Preparation of Mechanically Stable Superamphiphobic Coatings via Combining Phase Separation of Adhesive and Fluorinated SiO_2_ for Anti-Icing

**DOI:** 10.3390/nano13121872

**Published:** 2023-06-16

**Authors:** Jinfei Wei, Weidong Liang, Junping Zhang

**Affiliations:** 1Department of Chemical Engineering, College of Petrochemical Engineering, Lanzhou University of Technology, Lanzhou 730050, China; jinfeiwei2022@163.com; 2Center of Eco-Material and Green Chemistry, Lanzhou Institute of Chemical Physics, Chinese Academy of Sciences, Lanzhou 730000, China

**Keywords:** superamphiphobicity, mechanical stability, anti-icing, phase separation, adhesive

## Abstract

Superamphiphobic coatings have widespread application potential in various fields, e.g., anti-icing, anti-corrosion and self-cleaning, but are seriously limited by poor mechanical stability. Here, mechanically stable superamphiphobic coatings were fabricated by spraying the suspension composed of phase-separated silicone-modified polyester (SPET) adhesive microspheres with fluorinated silica (FD-POS@SiO_2_) on them. The effects of non-solvent and SPET adhesive contents on the superamphiphobicity and mechanical stability of the coatings were studied. Due to the phase separation of SPET and the FD-POS@SiO_2_ nanoparticles, the coatings present a multi-scale micro-/nanostructure. Combined with the FD-POS@SiO_2_ nanoparticles of low surface energy, the coatings present outstanding static and dynamic superamphiphobicity. Meanwhile, the coatings present outstanding mechanical stability due to the adhesion effect of SPET. In addition, the coatings present outstanding chemical and thermal stability. Moreover, the coatings can obviously delay the water freezing time and decrease the icing adhesion strength. We trust that the superamphiphobic coatings have widespread application potential in the anti-icing field.

## 1. Introduction

The adhesion and accumulation of ice can result in large damage to the infrastructure, and lead to serious problems such as communication interruption, traffic delays and power outages [1,2,3]. Therefore, the study of anti-icing materials and techniques has pulled widespread concentration [4,5,6,7,8,9,10]. In the past decades, there are two main approaches to solving the problem of icing: active anti-icing and passive anti-icing. Active anti-icing methods including thermal melting and mechanical vibration are traditional de-icing methods [10,11]. These methods often need massive manpower and resources, and, thus, result in high costs. Differently, the passive anti-icing methods including anti-icing coatings and anti-freeze proteins have many advantages such as low cost and energy consumption [12,13,14].

The micro-/nanostructures of superhydrophobic coatings can capture air pockets at the solid–liquid interfaces and can greatly decrease the solid–liquid contact area. Thus, superhydrophobic coatings can effectively prevent heat transfer and are promising passive anti-icing coatings [15,16,17,18,19]. However, superhydrophobic coatings are often exposed to the outdoor environment for anti-icing, where the coatings often contact with liquids of lower surface tension and containments in the air rather than pure water. Thus, superhydrophobic coatings often have short service life for outdoor applications including anti-icing [20]. Compared with superhydrophobic coatings, superamphiphobic coatings have lower surface energy and are very promising to solve this problem [21], but the poor mechanical stability still seriously limits their practical applications [22,23].

So far, the strategies to enhance the mechanical stability of superhydrophobic and superamphiphobic coatings mainly include: (1) protecting nanostructures using micro-skeletons [24,25]; (2) constructing self-similar structures [26,27]; (3) self-healing [28,29,30,31]; and (4) using adhesives [32,33,34], etc. Among these strategies, using adhesives has been widely favored by researchers due to the advantages of good performance, simple operation and low cost [34,35]. Zhang et al. prepared a skin-inspired triple-layered superhydrophobic coating using fluorinated epoxy resin, SiO_2_ and cellulose, which showed excellent mechano–chemical–thermal robustness [35]. Peng et al. reported a robust superhydrophobic coating by a combination of fluorinated epoxy resin, perfluoropolyether and polytetrafluoroethylene [36]. However, although adhesives can enhance the mechanical stability of the coatings, they also tend to embed low surface energy nanoparticles, resulting in the weakening of multi-scale micro-/nanostructures and a sharp increase of surface energy [37,38]. Therefore, it is very challenging to prepare mechanically stable superamphiphobic coating by introducing adhesives, because superamphiphobic coatings require more obvious multi-scale micro-/nanostructure and lower surface energy compared with superhydrophobic coatings [22].

Here, the mechanically stable superamphiphobic coatings were prepared by combining phase separation of a silicone-modified polyester (SPET) adhesive and fluorinated SiO_2_ (FD-POS@SiO_2_) nanoparticles for efficient anti-icing. First, a uniform suspension containing SPET/FD-POS@SiO_2_ microparticles was fabricated via non-solvent-induced phase separation of the suspension composed of SPET and the FD-POS@SiO_2_ nanoparticles. Then, the suspension was sprayed on the Al alloy plates and cured to obtain the mechanically stable superamphiphobic coatings. The phase separation of the SPET adhesive formed microspheres, and the FD-POS@SiO_2_ nanoparticles were wrapped on the microspheres to form the SPET/FD-POS@SiO_2_ microparticles with micro-/nanostructures and low surface energy. Therefore, the coatings present outstanding static and dynamic superamphiphobicity. The coatings also present outstanding mechanical stability, chemical stability and thermal stability. Furthermore, the coatings show good anti-icing performance.

## 2. Materials and Methods

### 2.1. Materials

SiO_2_ (10–20 nm) nanoparticles were bought from Maikun Chemical Co., Ltd., (Shanghai, China). Tetraethoxysilane (TEOS) and *1H*,*1H*,*2H*,*2H*-perfluorodecyltriethoxysilane (PFDTES) were bought from Gelest (Morrisville, NC, USA). The SPET adhesive was bought from Shandong Xinna Superhydrophobic New Material Co., Ltd., (Yantai, China). The Al alloy plates were bought from the Haocheng flagship store (Shanghai, China). The Al alloy plates were polished with 400# sandpapers, washed with acetone, ethanol and deionized water by ultrasonication, and finally dried in air. Hydroxyl-terminated polybutadiene (HTPB, average molecular weight = 3000) was supplied by China Haohua Chemical Group Co., Ltd. (Beijing, China). 1,1,1-Trichlorotrifluorocthane (CFC-113a), butyl acetate, dioctyl sebacate (DOS), and other reagents were supplied by China National Medicines Co., Ltd. (Beijing, China). The HTPB-H was obtained by mixing HTPB and DOS (1:1). All the reagents were used as received without further purification.

### 2.2. Preparation of FD-POS@SiO_2_ Nanoparticles

The FD-POS@SiO_2_ nanoparticles were fabricated according to our previous work [32,33,34]. An amount of 100 g of SiO_2_ nanoparticles was dispersed into 5 L of ethanol containing 0.4 L of ammonia solution by mechanical stirring and sonication. Subsequently, 150 mL of PFDTES and TEOS were added to the suspension. After a 2 h reaction at room conditions, the low surface energy FD-POS@ SiO_2_ nanoparticles suspension was obtained. Then, the suspension was washed using butyl acetate 3 times. Finally, the semi-solid FD-POS@silica nanoparticles containing butyl acetate were obtained by centrifugation.

### 2.3. Preparation of SPET/FD-POS@SiO_2_ Coatings

First, 2.4 g of SPET was dissolved in 7.9 mL of butyl acetate. Then, 4.9 g of semi-solid FD-POS@SiO_2_ nanoparticles containing butyl acetate were dispersed into the SPET solution under continuous stirring. After stirring for 1 h, 1.7 mL of CFC-113a was added dropwise to induce phase separation of the SPET adhesive. Finally, a uniform suspension containing the SPET/FD-POS@SiO_2_ microparticles was prepared by continuous stirring for 3 h. Subsequently, the SPET/FD-POS@SiO_2_ superamphiphobic coatings were fabricated by spraying 4 mL of the suspension onto 12 cm^2^ polished Al alloy plates and curing at 275 °C for 15 min. The coatings in other sizes were prepared according to the same procedure.

### 2.4. Static and Dynamic Superamphiphobicity Tests

The static superamphiphobicity of the coatings was studied by testing the contact angle (CA) and sliding angle (SA) of 10 μL droplets (water, HTPB-H, soybean oil and *n*-decane). The CA and SA were measured by the contact angle system OCA20 (Dataphysics, Filderstadt, Germany), and a minimum of five positions were measured.

The dynamic superamphiphobicity of the coatings was studied by impacting droplets (10 μL) from a certain height on the coating surface inclined at 45°. The maximum droplet release height that the coating can resist without any adhesion was defined as the maximum release height. The higher the maximum release height, the better the dynamic superamphiphobicity.

### 2.5. Stability Tests

The mechanical stability of the coatings was studied using the Taber abrasion test (ASTM D4060) and the tape peeling test. For the Taber abrasion test, a Taber-type abrasion tester (Dongguan Yaoke Instrument Equipment Co., Ltd., Dongguan, China) with CS 10 grinding wheel was employed and the load was 125 g. After certain cycles, the CA_soybean oil_ and SA_soybean oil_ were tested. For the tape peeling test, the 3M tape (20 mm in width and 0.6 mm in thickness) with an adhesion strength of 3000 N m^−1^ to standard stainless steel was used. The 3M tape was stuck to the surface of the coating using a cylindrical copper block (2.3 kPa). Then, the tape was peeled off, and the CA_soybean oil_ and SA_soybean oil_ of the coatings were measured.

The chemical stability of the coatings was studied by immersion in 1 M HCl(aq), 1 M NaOH(aq) and 1 M NaCl(aq) solutions for some time. At certain time intervals, the CA_soybean oil_ and SA_soybean oil_ were measured.

The thermal stability of the coatings was studied by high-/low-temperature treatment. The coating was placed horizontally in a bake oven at 150 °C or in a refrigerator at −18 °C for a period of time. At certain time intervals, the CA_soybean oil_ and SA_soybean oil_ were measured.

### 2.6. Anti-Icing Tests

The anti-icing performance of the coatings was studied via the use of self-constructed ice adhesion devices. The temperature in the devices was controlled at 2~3 °C via a constant temperature water bath (−5 °C), and the RH was controlled at 60% via a mixture of wet and dry N_2_. The coatings were placed horizontally on a cooling plate at −5 °C. Then, the freezing process of water (60 µL deionized water) was recorded using a CCD camera to measure the water freezing time. The ice adhesion strength test was conducted by injecting 1.0 mL of deionized water into a glass column on the surface of the coating and completely freezing after 3 h. Then, remove the icicle and record the force (F) that separates the icicle from the coating. The contact area (A) between the icicle and the coating surface is 100 mm^2^. Therefore, the ice adhesion strength (τ) can be calculated by Equation (1).
τ = F/A(1)

### 2.7. Characterization

A field emission scanning electron microscope (SEM, JSM-6701F, JEOL) was employed to study the morphology of the coatings. Before SEM observation, the samples were fixed on copper stubs using conductive tape and coated with a layer of gold film (ca. 7 nm in thickness). The Fourier transform infrared (FT-IR) spectrum of the SPET/FD-POS@SiO_2_ was recorded on the Thermo Nicolet Nexus spectrophotometer (Thermo, Madison, WI, USA) in 4000–400 cm^−1^ using KBr pellets. The thicknesses of the coatings were tested by an electronic digital display micrometer with a resolution of 1 μm (SYA1704569, SYNTEK). The thicknesses of the coatings were calculated by Equation (2).
Coating thickness = T_1_ − T_2_(2)
where T_1_ is the thickness of the Al alloy plate with the coating and T_2_ is the thickness of the Al alloy plate.

## 3. Results and Discussion

### 3.1. Preparation of SPET/FD-POS@SiO_2_ Superamphiphobic Coatings

Figure 1a shows the schematic preparation of the SPET/FD-POS@SiO_2_ coatings via non-solvent-induced phase separation. First, a proper amount of the SPET adhesive was dissolved in butyl acetate. Then, the FD-POS@SiO_2_ nanoparticles were dispersed into the SPET solution in butyl acetate by magnetic stirring. Next, CFC-113a, the non-solvent of the SPET adhesive, was added dropwise with magnetic stirring to trigger phase separation of the SPET adhesive. Finally, the uniform SPET/FD-POS@SiO_2_ dispersion liquid was formed by stirring (Appendix A). The SPET/FD-POS@SiO_2_ superamphiphobic coatings were fabricated via spray-coating the SPET/FD-POS@SiO_2_ dispersion liquid onto the Al alloy plates followed by curing at 275 °C.

The surface morphology of the Al alloy plate and SPET/FD-POS@SiO_2_-coated Al alloy plate was studied by SEM (Figure 1b,c). The surface of the Al alloy plate was relatively smooth with small scratches due to 400# sandpaper polishing (Appendix A). After spraying with the SPET/FD-POS@SiO_2_ coating, the surface was very rough with numerous microspheres and their microaggregates. The microspheres are composed of the SPET core and the FD-POS@SiO_2_ nanoparticles shell with a diameter of 2–4 μm. At high magnification, the microspheres and microaggregates show multi-scale micro-/nanostructure. This is attributed to the phase separation and adhesion effect of SPET [32,33,34].

The chemical composition of the SPET/FD-POS@SiO_2_-coating was analyzed by FT-IR spectroscopy (Figure 1d). The peak at 3438 cm^−1^ is due to the residual -OH groups [39]. The peak at 1102, 810 and 468 cm^−1^ are attributed to the Si-O-Si groups [39]. Furthermore, the peaks corresponding to the C-F groups were detected at 1238 and 1146 cm^−1^. This means the presence of -Si(CH_2_)_2_(CF_2_)_7_CF_3_ groups on the surface of the coating, which could effectively reduce the surface energy and, thus, endow the coating with excellent superamphiphobicity [21]. Moreover, the peak at 1732 cm^−1^ is attributed to the O-C = O groups [40,41], which mainly originate from the SPET adhesive.

### 3.2. Effects of Phase Separation Induced by Non-Solvent

The non-solvent (CFC-113a) content determines the phase separation degree of the SPET adhesive and thus affects the state of SPET in the SPET/FD-POS@SiO_2_ suspension and comprehensive performance of the SPET/FD-POS@SiO_2_ coatings, such as static and dynamic superamphiphobicity and mechanical stability. Therefore, the effects of the non-solvent content on the coating performance were studied systematically. As an example of complicated liquids, HTPB-H with a surface tension of 36.2 mN m^−1^ (20 °C) was used as the probing liquid. HTPB-H is a widely used adhesive in various fields.

As can be seen from Appendix A, when the non-solvent content was 1.0 mL, the SPET/FD-POS@SiO_2_ coating was very flat with obvious cracks. This is because phase separation of the SPET adhesive did not occur at such low non-solvent content. Further increase in the non-solvent content and phase separation of the SPET adhesive occurred, generating rougher SPET/FD-POS@SiO_2_ coatings. When the non-solvent content was 2.7 mL, obvious aggregates appeared on the coating surface. This is mainly because the non-solvent content was in excess, which resulted in the precipitation of the SPET adhesive. Subsequently, the static superamphiphobicity of the SPET/FD-POS@SiO_2_ coatings was studied by CA_HTPB-H_ and SA_HTPB-H_ (Figure 2a). When the non-solvent content was less than 1.7 mL, the SPET/FD-POS@SiO_2_ coatings showed similar static superamphiphobicity (CA_HTPB-H_ = 155°, SA_HTPB-H_ = 4°). With an increase in the non-solvent content, the static superamphiphobicity gradually decreased. When the non-solvent content was 2.7 mL, the CA_HTPB-H_ and SA_HTPB-H_ of the SPET/FD-POS@SiO_2_ coating were 152° and 9.8°, respectively.

The non-solvent content affects the dynamic superamphiphobicity of the SPET/FD-POS@SiO_2_ coatings (Figure 2b). The dynamic superamphiphobicity was studied by impacting droplets (10 μL) from a certain height on the coating surface inclined at 45°. The maximum droplet release height that the coating can resist without any adhesion was defined as the maximum release height. The higher the maximum release height, the better the dynamic superamphiphobicity [33]. When the non-solvent content was 1.0 mL, the maximum release height that the coating can resist HTPB-H adhesion was 11 cm. When the non-solvent content increased to 1.7 mL, the maximum release height increased to 15 cm. This is mainly because the coating has obvious multi-scale micro-/nanostructure and low surface energy (Figure 1b,c). However, with the further increase of the non-solvent content, the maximum release height of HTPB-H gradually decreased, which is mainly because the excessive non-solvent led to serious precipitation of SPET.

The non-solvent content also affects the mechanical stability of the SPET/FD-POS@SiO_2_ coatings (Figure 2c and Appendix A). Since the SPET/FD-POS@SiO_2_ coating with a non-solvent content of 1.0 mL has obvious cracks, its mechanical stability was not tested. When the non-solvent content was 1.4 mL, the SPET/FD-POS@SiO_2_ coating had the best mechanical stability and could withstand 40 cycles of Taber abrasion. The SPET/FD-POS@SiO_2_ coatings with non-solvent contents of 1.7 and 2.2 mL can withstand 30 cycles of Taber abrasion. However, the coating thickness with a non-solvent content of 2.2 mL was reduced faster during Taber abrasion (Figure 2d). Further increase in the non-solvent content to 2.7 mL resulted in a further decline of the mechanical stability, which could only withstand 20 cycles of Taber abrasion. Considering both dynamic superamphiphobicity and mechanical stability, the SPET/FD-POS@SiO_2_ coating with 1.7 mL non-solvent was used for further studies.

### 3.3. Effects of SPET Adhesive Content

The effects of the SPET adhesive content on the coating performance were studied systematically (Figure 3). As the SPET adhesive content decreased from 3.0 g to 1.8 g, the CA_HTPB-H_ of the SPET/FD-POS@SiO_2_ coating did not exhibit obvious change, but the SA_HTPB-H_ decreased slightly from 4° to 2.5° (Figure 3a). This is because the surface energy of the SPET adhesive is high compared to the FD-POS@SiO_2_ nanoparticles, and thus more SPET adhesive increased the surface energy of the coatings. In addition, the SPET adhesive content also affects the dynamic superamphiphobicity of the SPET/FD-POS@SiO_2_ coatings (Figure 3b). When the SPET adhesive content was 2.4 g, the maximum release height of the coating can resist HTPB-H adhesion is 18 cm. With the further reduction of the SPET adhesive content, the maximum release height will further increase.

The SPET adhesive content also affects the mechanical stability of the SPET/FD-POS@SiO_2_ coatings (Figure 3c). With the decrease of the SPET adhesive content, the mechanical stability of the coatings first increased and then decreased. In addition, the thickness of the SPET/FD-POS@SiO_2_ coatings decreased faster during the Taber abrasion and the wear mark were more obvious (Figure 3d and Appendix A). When the SPET adhesive content was 2.4 g, the SPET/FD-POS@SiO_2_ coating had the best mechanical stability. After 50 cycles of Taber abrasion, the SPET/FD-POS@SiO_2_ coating was still superamphiphobic (CA_HTPB-H_ = 150°, SA_HTPB-H_ = 38°). Considering both dynamic superamphiphobicity and mechanical stability, the SPET/FD-POS@SiO_2_ coating with 2.4 g SPET adhesive was used for further studies.

### 3.4. Static and Dynamic Superamphiphobicity of SPET/FD-POS@SiO_2_ Coatings

The Al alloy plate is amphiphilic for various liquids (e.g., water, HTPB-H, soybean oil and *n*-decane) (Appendix A). In contrast, after spraying with the SPET/FD-POS@SiO_2_ coating, the surface showed excellent static superamphiphobicity for liquids with surface tension higher than 23.8 mN m^−1^ (*n*-decane) (Figure 4a,b). Water, HTPB-H, soybean oil and *n*-decane droplets are spherical on the surface of the SPET/FD-POS@SiO_2_ coating and could roll off the inclined coatings. When immersed in water and soybean oil, strong light reflection can be observed on the surface of the SPET/FD-POS@SiO_2_ coatings (Figure 4c). The coatings could remain dry without an obvious change in superamphiphobicity after being immersed in water and soybean oil for at least 24 h (Appendix A). These results proved the existence of a stable air cushion at the interface of the SPET/FD-POS@SiO_2_ coating and the liquids are in the Cassie–Baxter state [29]. In addition, the coatings showed good self-cleaning performance due to the excellent superamphiphobicity (Appendix A).

The dynamic superamphiphobicity of the SPET/FD-POS@SiO_2_ coating was studied by impacting droplets (10 μL) from a certain height on the coating surface inclined at 45° (Figure 4d). The maximum release heights of water, HTPB-H, soybean and *n*-decane are 162, 18, 28 and 3.8 cm, respectively, which proves that the coating has good dynamic superamphiphobicity. This is mainly attributed to both multi-scale micro-/nanostructure and the low surface energy of the coating.

### 3.5. Stability of SPET/FD-POS@SiO_2_ Superamphiphobic Coatings

The mechanical stability of the SPET/FD-POS@SiO_2_ coating was studied by the Taber abrasion test and tape-peeling test (Figure 5a,b). In the Taber abrasion test, CA_soybean oil_ decreased to 150.3° after 10 abrasion cycles and SA_soybean oil_ increased to 12° and then remained unchanged after 20 abrasion cycles (Figure 5a). Even after 50 abrasion cycles, the SPET/FD-POS@SiO_2_ coating still kept good superamphiphobicity (CA_soybean oil_ = 150°, SA_soybean oil_ = 34.6°). The mechanical stability of the coating was further studied by the tape-peeling test. The CA_soybean oil_ decreased to 150.1° and SA_soybean oil_ increased to 15.0° after 25 peeling cycles and remained unchanged after 50 cycles (Figure 5b). After 100 cycles, the coating was still superamphiphobic (CA_soybean oil_ = 150°, SA_soybean oil_ = 32.6°). These results confirmed that the SPET/FD-POS@SiO_2_ coating has outstanding mechanical stability, which is attributed to the adhesion effect of SPET [34,35,36].

In addition, the chemical stability of the SPET/FD-POS@SiO_2_ coating was studied via the immersion test in corrosive liquids (Figure 5c). After immersing in 1 M HCl_(aq)_ and 1 M NaCl_(aq)_ for 4 h, the superamphiphobicity of the coating has no obvious change. When immersed in 1 M NaOH_(aq)_, the superamphiphobicity decreased slightly but was still superamphiphobic after 4 h. Moreover, the coating showed excellent repellence to concentrated H_2_SO_4_ and saturated NaOH_(aq)_ (CA_Concentrated H2SO4_ = 160.8°, SA_Concentrated H2SO4_ = 2.8°; CA_Saturated NaOH_(aq)__ = 163.5°, SA_Saturated NaOH_(aq)__ = 1.5°) (Appendix A). These results confirmed that the SPET/FD-POS@SiO_2_ coating has outstanding chemical stability. This is mainly because the main component of the acid and the base is water with a small amount of solute. So, their surface tension is close to that of water. Therefore, the coating has a very low contact area with these liquids such as with water. Moreover, the chemical inertness of SPET and FD-POS@SiO_2_ endows the coating with good chemical stability.

Furthermore, the thermal stability of the SPET/FD-POS@SiO_2_ coating was studied via high-/low-temperature treatment (Figure 5d). After treatment at 150 °C or −18 °C for 4 h, the superamphiphobicity of the coating had no obvious change, which demonstrates excellent thermal stability.

### 3.6. Anti-Icing Performance of SPET/FD-POS@SiO_2_ Superamphiphobic Coatings

The anti-icing performance of various materials is often evaluated by testing the water freezing time and the ice adhesion strength [42,43]. The freezing time of methylene blue stained water droplets (60 μL) on the Al alloy plate and the SPET/FD-POS@SiO_2_ coated Al alloy plate was measured at −5 °C and 60% relative humidity. On the Al alloy plate, the water droplets completely froze after 94.3 ± 3.3 s. Compared with the Al alloy plate, the icing time of water droplets on the SPET/FD-POS@SiO_2_-coated Al alloy plates was delayed to 224.0 ± 7.3 s, i.e., ~2.4 times (Figure 6a–c, Appendix A). Subsequently, the anti-icing performance of the SPET/FD-POS@SiO_2_ coating was further evaluated via the ice adhesion strength test. On the Al alloy plate, the ice adhesion strength was 247.3 ± 11.6 kPa (Figure 6d). In contrast, for the SPET/FD-POS@SiO_2_-coated Al alloy plate, the ice adhesion strength decreased to 77.3 ± 11.2 kPa (Figure 6d). These results proved that the SPET/FD-POS@SiO_2_ coating has good anti-icing performance. This is because the SPET/FD-POS@SiO_2_ coating trapped a steady air layer at the solid–water interface, which enormously reduced the contact area and inhibited heat transfer between the coating and water, and weakened the ice adhesion strength on the coating [32,44].

## 4. Conclusions

In summary, mechanically stable superamphiphobic coatings were fabricated by spraying the suspension composed of phase-separated SPET adhesive microspheres with FD-POS@SiO_2_ nanoparticles on them. The non-solvent and SPET contents greatly affect static and dynamic superamphiphobicity as well as the mechanical stability of the coatings. After systematic optimization, the coatings show outstanding static and dynamic superamphiphobicity. The coatings have high contact angle, low sliding angle and high maximum release height for various liquids with surface tension not less than 23.8 mN m^−1^_._ The coatings also show excellent mechanical, chemical and thermal stability. Meanwhile, the coatings have good anti-icing performance, e.g., long water freezing time and low ice adhesion strength. Therefore, we anticipate that the coatings may be applied in the anti-icing field.

## Figures and Tables

**Figure 1 nanomaterials-13-01872-f001:**
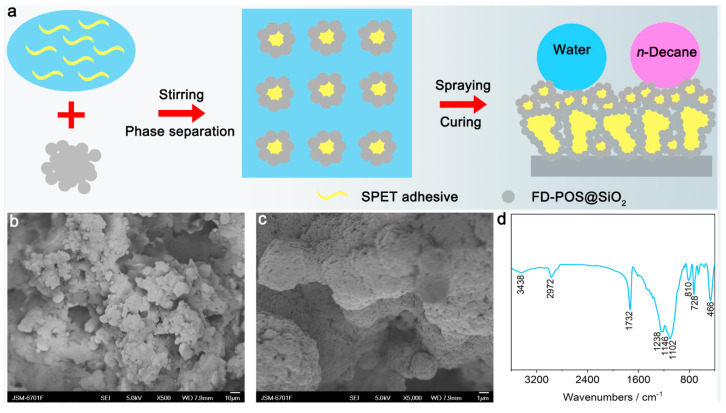
(**a**) Schematic preparation of the SPET/FD-POS@SiO_2_ coatings. (**b**,**c**) SEM images and (**d**) FT-IR spectrum of the SPET/FD-POS@SiO_2_ superamphiphobic coating. V_non-solvent_ = 1.7 mL, m_SPET_ = 2.4 g.

**Figure 2 nanomaterials-13-01872-f002:**
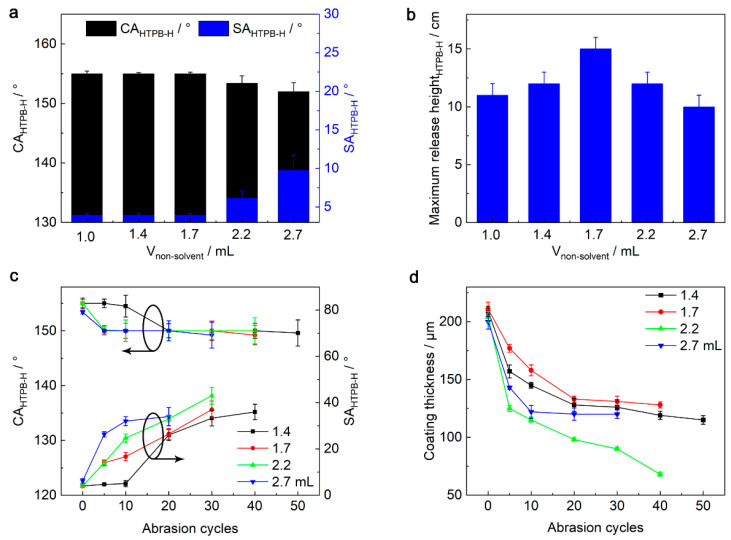
(**a**) Changes of CA_HTPB-H_ and SA_HTPB-H_ and (**b**) the maximum release height of HTPB-H droplets with the non-solvent content during preparation of the SPET/FD-POS@SiO_2_ coatings. Changes of (**c**) CA_HTPB-H_ and SA_HTPB-H_ and (**d**) coating thickness of the SPET/FD-POS@SiO_2_ coatings with different non-solvent content in the Taber abrasion test. m_SPET_ = 3.0 g.

**Figure 3 nanomaterials-13-01872-f003:**
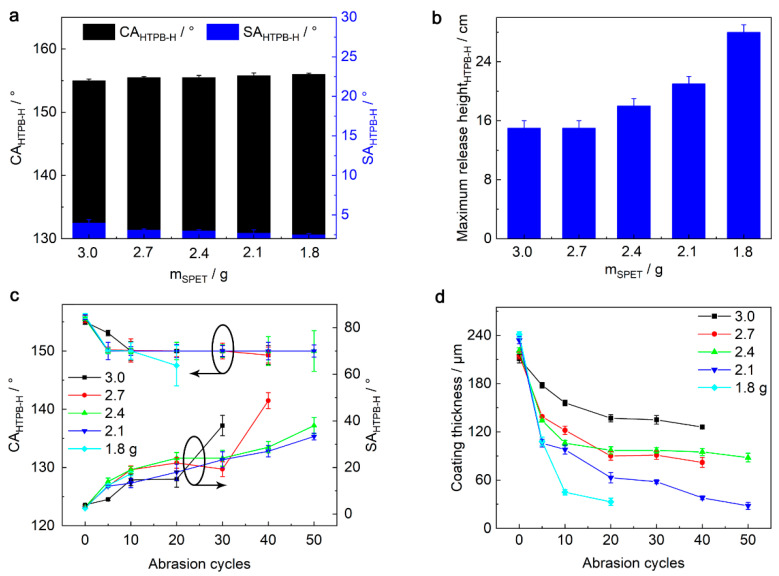
(**a**) Changes of CA_HTPB-H_ and SA_HTPB-H_ and (**b**) the maximum release height of HTPB-H droplets with the SPET adhesive content during preparation of the SPET/FD-POS@SiO_2_ coatings. Changes of (**c**) CA_HTPB-H_ and SA_HTPB-H_ and (**d**) coating thickness of the SPET/FD-POS@SiO_2_ coatings with different SPET contents during Taber abrasion. V_non-solvent_ = 1.7 mL.

**Figure 4 nanomaterials-13-01872-f004:**
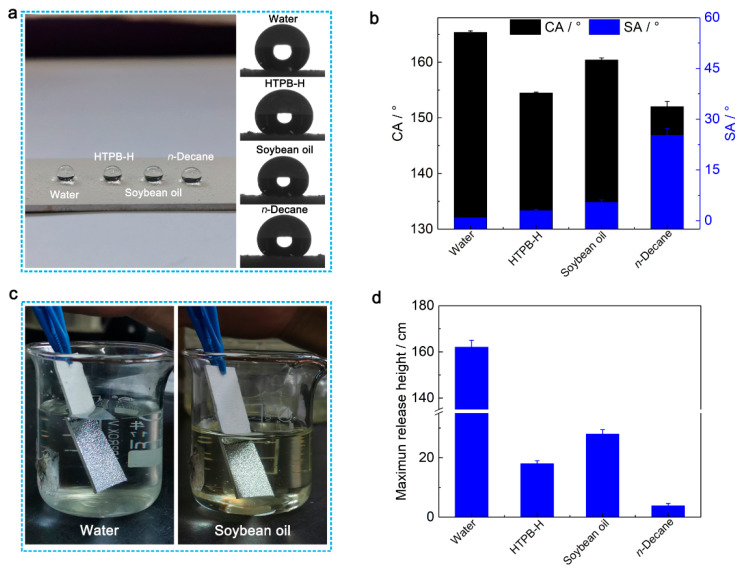
(**a**) Photographs of the SPET/FD-POS@SiO_2_ coating with droplets of different surface tension on the surface. (**b**) CA and SA of various liquids on the SPET/FD-POS@SiO_2_ coating. (**c**) Photographs of the SPET/FD-POS@SiO_2_ coatings immersed in water and soybean oil. (**d**) The maximum release height of various liquids on the SPET/FD-POS@SiO_2_ coating. V_non-solvent_ = 1.7 mL, m_SPET_ = 2.4 g.

**Figure 5 nanomaterials-13-01872-f005:**
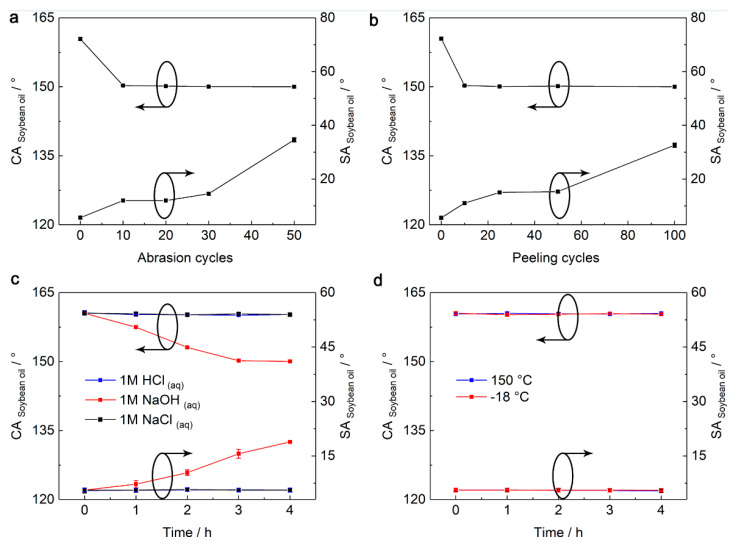
Changes of CA_soybean oil_ and SA_soybean oil_ of the SPET/FD-POS@SiO_2_ coating during the (**a**) Taber abrasion test, (**b**) tape-peeling test, (**c**) corrosive liquids immersion tests and (**d**) high-/low-temperature treatment tests. V_non-solvent_ = 1.7 mL, m_SPET_ = 2.4 g.

**Figure 6 nanomaterials-13-01872-f006:**
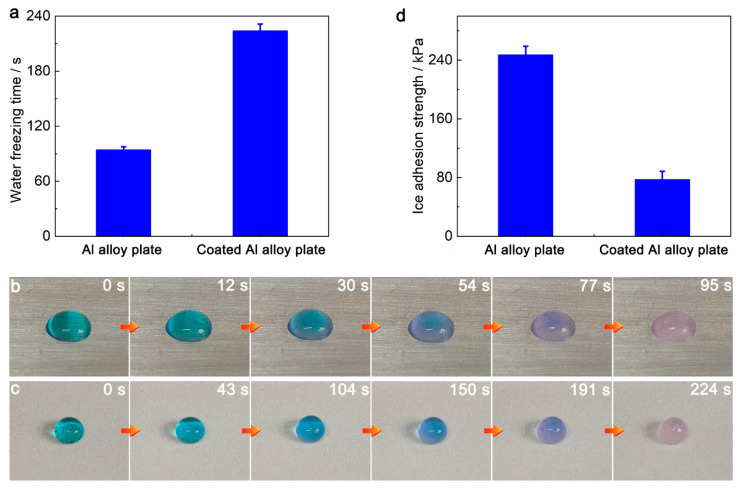
(**a**) Freezing time and (**b**,**c**) freezing process (−5 °C) of water droplets (60 µL) on the Al alloy plate and SPET/FD-POS@SiO_2_-coated Al alloy plate. (**d**) Ice adhesion strength on the Al alloy plate and SPET/FD-POS@SiO_2_-coated Al alloy plate. V_non-solvent_ = 1.7 mL, m_SPET_ = 2.4 g.

## Data Availability

The data presented in this study are available on request from the corresponding author.

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
