# Peer review of "Preparation of Mechanically Stable Superamphiphobic Coatings via Combining Phase Separation of Adhesive and Fluorinated SiO2 for Anti-Icing"

_nanomaterials, 2023, doi:10.3390/nano13121872_

Round 1

Author Response

1) A brief summary:

The article “Preparation of mechanically stable superamphiphobic coatings via combining phase separation of adhesive and fluorinated SiO2 for Anti-icing” shows great results not only in anti-acing fields but also excellent chemical and thermal stability due to physical repelling of various liquids. The formulation solved the main problematic issue around the superamphiphobicity-a mechanical stability. The mechanical stability was checked by Taber abrasion test, and adhesion strength was verified by tape peeling test.

Reply: Thank you for your recognition.

2) General concept comments:

2.1) The article lacks a comparative analysis between an untreated surface and a surface coated with a given formulation. There is only anti-icing effect comparison between treated and untreated plates. It is necessary to add a SEM micrograph of the aluminum surface itself as well as contact angle and sliding angle for Al plate in the graphs as control samples.

Reply: Thank you for the insightful advice. The SEM images of the Al alloy plate has been supplemented in the revised manuscript. The surface of Al alloy plate was relatively smooth with small scratches due to 400# sandpaper polishing (Fig. S2). In addition, photographs of the Al alloy plate with droplets of different surface tension on the surface and CA of various liquids on the Al alloy plate (Fig. S6) were supplemented in the revised manuscript.

2.2) There is also a lack of a description of the resulting coating morphology: are the microspheres hollow, thickness, etc.

Reply: The microspheres are composed of the SPET core and the FD-POS@SiO2 nanoparticles shell with a diameter of 2-4 μm. The description has been supplemented in the revised manuscript.

2.3) It should be indicated the FTIR analysis purpose and what hypothesis it confirms. How the resulting chemical molecules affects the superamphiphobicity.

Reply: Thank you for the insightful advice. The FTIR analysis purpose is to study the chemical composition of the SPET/FD-POS@SiO2 coating. The peaks corresponding to the C-F groups were detected at 1238 and 1146 cm−1. This means presence of -Si(CH2)2(CF2)7CF3 groups on the surface of the coating, which could effectively reduce the surface energy and thus endow the coating with excellent superhydrophobicity. The description has been supplemented in the revised manuscript.

2.4) In some cases, there is a lack of scientific support for the stated facts like: Why exactly the coatings repel the acid and the base? Is this due to morphological structure only or any chemical process occurs there?

Reply: There are two reasons why coatings can repel the acid and the base. Firstly, the main component of the acid and the base was water with a small amount of solute. So, their surface tension was close to that of water. Therefore, the coating has very low contact area with these liquids like with water. Moreover, the chemical inertness of SPET and FD-POS@SiO2 endows the coating with good chemical stability. Therefore, the SPET/FD-POS@SiO2 coating can repel the acid and the base. The description has been supplemented in the revised manuscript.

2.5) If there was checked the maximum acid and base concentration that the coating can stand?

Reply: The maximum acid and base concentration that the coating can withstand has been tested according to your suggestion. The coating show excellent repellence to concentrated H2SO4 and saturated NaOH(aq) (CAconcentrated H2SO4 = 160.8°, SAconcentrated H2SO4 = 2.8°; CAsaturated NaOH(aq) = 163.5°, SAsaturated NaOH(aq) = 1.5°) (Fig. S9). The data has been supplemented in the revised manuscript.

2.6) For how long the plates were immersed in water and soybean oil and how it changes the coating.

Reply: The coatings remained dry without obviously change in superamphiphobicity after being immersed in water and soybean oil for at least 24 h, respectively (Fig. S7). The data has been supplemented in the revised manuscript.

3) Specific comments:

3.1) Line 36: no need “etc.” word

Reply: The “etc.” has been deleted in the revised manuscript.

3.2) Line 147: typo “A” to “a”

Reply: The “A” has been corrected in the revised manuscript.

3.3) Line 149: typo “magnetic stirring for”

Reply: The “magnetic stirring for” has been corrected in the revised manuscript.

3.4) Line 72-74: to much conjunctive adverbs in a row like: therefore, moreover, furthermore.

Reply: This has been revised in the revised manuscript.

3.5) Line 200 and line 237: figure’s title should be consistent and identical to another figure (starts from (a) “….”)

Reply: '(a)' has been moved to the foremost in the revised manuscript.

3.6) Line 253: figure 4 (a) All the liquid’s names should be written the same: above the droplet or inside.

Reply: All the liquid’s names has been written above the droplet in the revised manuscript.

3.7) Line 267: need to add word “then”: “to 12 and …then…remained.

Reply: “then” has been added in the revised manuscript.

3.8) Line 286: no need the sentence: “This is because of excellent thermal stability” because it has been mentioned before.

Reply: The sentence has been deleted in the revised manuscript.

3.9) Line 321: rephrase: may be applied in the anti-acing field.

Reply: The sentence has been modified in the revised manuscript.

Reviewer 2 Report

The research topic presented in the peer-reviewed manuscript concerns superamphiphobic coatings for application in the anti-icing field. Mechanically stable superamphiphobic coatings were prepared by spraying on substrate the suspension composed of phase-separated silicone modified polyester (SPET) and fluorinated silica (FD-POS@SiO2) nanoparticles. The authors present the method of preparation of coatings and methods of their testing. The subjects of the research were: static (contact angle, CA) and dynamic (sliding angkle, SA) superamphiphobicity, thermal and chemical stability as well as anti-icing. In addition, the morphology and chemical composition of the SPET/FD-POS@SiO2 coating was studied. The mechanical stability of the produced layers was also tested.

It was established, that due to phase separation of SPET and the FD-POS@SiO2 nanoparticles, the coatings present a hierarchical micro-/nanostructure thanks to which it presents excellent superam-phiphobicity. Moreover, according to the authors, the elaborated coatings also show excellent mechanical, chemical and thermal stability.

The peer-reviewed manuscript is written in a clear and understandable way. The results of the research are presented in a legible manner and I consider their discussion to be factually correct. In my opinion, it contains valuable results that may be of interest to many potential readers.

I counted 16 self-citations. In my opinion, this is too much and I consider such a practice to be inappropriate. So I think authors should greatly reduce the number of self-citations. With this remark in mind, I believe that the peer-reviewed paper will be suitable for publication in Nanomaterials.

Author Response

The research topic presented in the peer-reviewed manuscript concerns superamphiphobic coatings for application in the anti-icing field. Mechanically stable superamphiphobic coatings were prepared by spraying on substrate the suspension composed of phase-separated silicone modified polyester (SPET) and fluorinated silica (FD-POS@SiO2) nanoparticles. The authors present the method of preparation of coatings and methods of their testing. The subjects of the research were: static (contact angle, CA) and dynamic (sliding angkle, SA) superamphiphobicity, thermal and chemical stability as well as anti-icing. In addition, the morphology and chemical composition of the SPET/FD-POS@SiO2 coating was studied. The mechanical stability of the produced layers was also tested.

It was established, that due to phase separation of SPET and the FD-POS@SiO2 nanoparticles, the coatings present a hierarchical micro-/nanostructure thanks to which it presents excellent superamphiphobicity. Moreover, according to the authors, the elaborated coatings also show excellent mechanical, chemical and thermal stability.

The peer-reviewed manuscript is written in a clear and understandable way. The results of the research are presented in a legible manner and I consider their discussion to be factually correct. In my opinion, it contains valuable results that may be of interest to many potential readers.

I counted 16 self-citations. In my opinion, this is too much and I consider such a practice to be inappropriate. So I think authors should greatly reduce the number of self-citations. With this remark in mind, I believe that the peer-reviewed paper will be suitable for publication in Nanomaterials.

Reply: Thank you for your recognition. We have removed 9 self-citations in the revised manuscript.
